# Rapid Detection of Plasmid AmpC Beta-Lactamases by a Flow Cytometry Assay

**DOI:** 10.3390/antibiotics11081130

**Published:** 2022-08-19

**Authors:** Inês Martins-Oliveira, Blanca Pérez-Viso, Ana Silva-Dias, Rosário Gomes, Luísa Peixe, Ângela Novais, Rafael Cantón, Cidália Pina-Vaz

**Affiliations:** 1FASTinov: S.A., 4450-676 Matosinhos, Portugal; 2Division of Microbiology, Department of Pathology, Faculty of Medicine, University of Porto, 4200-319 Porto, Portugal; 3Servicio de Microbiología, Hospital Universitario Ramón y Cajal, Instituto Ramón y Cajal de Investigación Sanitaria (IRYCIS), 28034 Madrid, Spain; 4CIBER de Enfermedades Infecciosas, Instituto de Salud Carlos III, 28029 Madrid, Spain; 5CINTESIS-Center for Health Technology and Services Research, Faculty of Medicine, University of Porto, 4200-450 Porto, Portugal; 6UCIBIO-Applied Molecular Biosciences Unit, Department of Biological Sciences, Faculty of Pharmacy, University of Porto, 4050-313 Porto, Portugal; 7Associate Laboratory i4HB-Institute for Health and Bioeconomy, Faculty of Pharmacy, University of Porto, 4050-313 Porto, Portugal

**Keywords:** pAmpC, flow cytometry, resistance mechanisms, rapid antimicrobial susceptibility testing

## Abstract

Plasmidic AmpC (pAmpC) enzymes are responsible for the hydrolysis of extended-spectrum cephalosporins but they are not routinely investigated in many clinical laboratories. Phenotypic assays, currently the reference methods, are cumbersome and culture dependent. These methods compare the activity of cephalosporins with and without class C inhibitors and the results are provided in 24–48 h. Detection by molecular methods is quicker, but several genes should be investigated. A new assay for the rapid phenotypic detection of pAmpC enzymes of the *Enterobacterales* group-I (not usually AmpC producers) based on flow cytometry technology was developed and validated. The technology was evaluated in two sites: FASTinov, a spin-off of Porto University (Portugal) where the technology was developed, and the Microbiology Department of Ramón y Cajal University Hospital in Madrid (Spain). A total of 100 strains were phenotypically screened by disk diffusion for the pAmpC with the new 2 h assay. Molecular detection of the pAmpC genes was also performed on discrepant results. Forty-two percent of the strains were phenotypically classified as pAmpC producers using disk diffusion. The percentage of agreement of the flow cytometric assay was 93.0%, with 95.5% sensitivity and 91.1% specificity. Our proposed rapid assay based on flow cytometry technology can, in two hours, accurately detect pAmpC enzymes.

## 1. Introduction

Antimicrobial resistance (AMR) is a worldwide problem that requires global and rapid action and is a leading cause of death around the world. In 2019, it was estimated that 4.95 million deaths were associated with bacterial AMR [1]. Among resistance threats, multidrug-resistant (MDR) *Enterobacterales* are one of the most common causes of bacterial infections in both community and hospitalized patients, frequently resistant to third-generation cephalosporins. Resistance to these β-lactam antibiotics is mainly due to the production of the extended spectrum β-lactamases (ESBLs), the AmpC enzymes [2,3,4] and the most feared carbapenemases. β-lactamases are classified in four different molecular classes: A, B, C and D (Ambler classification) [5]. AmpC β-lactamases belong to Ambler class C that cause the hydrolysis of the β-lactam ring of penicillins, cephalosporins and monobactams. In general, they are poorly inhibited by the classical ESBL inhibitors, especially clavulanic acid [6]. They can be classified as chromosomal AmpCs (cAmpCs)—that might have inducible and hyperproduction expression—or plasmid AmpCs (pAmpCs) that, with some exceptions, are constitutively expressed. AmpC enzymes are frequently found as chromosomal β-lactamases in different species of Gram-negative bacteria [7]. Most species of *Enterobacter*, *Citrobacter* and *Serratia* (*Enterobacterales* group-II) are cAmpC producers while pAmpCs have been mainly described in the *Enterobacterales* group-I species (*Escherichia coli*, *Klebsiella pneumoniae*, *Proteus mirabilis*, *Salmonella* spp. and *Shigella* spp.) [8].

The detection of pAmpC β-lactamases is relevant for both epidemiological and clinical relevance as the isolates expressing these enzymes are often resistant to several antimicrobials, which significantly reduces the therapeutic options available and should be encouraged to avoid its undetected diffusion which might have negative impact on public health [9]. In addition, the treatment of infections due to the pAmpC-producer microorganisms is still under discussion [4,10]. Bacteria-producing pAmpCs have been described all over the world [2,11,12], with *bla*_CMY-2_ being the most frequent and globally distributed enzyme [8]. Nevertheless, epidemiological studies are scarce and their real prevalence might be underestimated due to the difficulties in laboratory diagnosis. The detection of pAmpC enzymes is not usually performed in most routine clinical microbiology laboratories. Unlike ESBLs, most of the automated systems do not provide specific detection of pAmpCs, and their presence might be masked by the concomitant production of ESBLs. EUCAST recommends confirming the presence of pAmpCs in the *Enterobacterales* group-I isolates, which are resistant to cefotaxime and/or ceftazidime and cefoxitin [13], using a cloxacillin synergy test in a growth-dependent assay, with at least a 24 h turnaround time. When a synergic effect is observed, it means that the strain is potentially a pAmpC producer. Nevertheless, a constitutive expression of the chromosomal AmpC β-lactamase cannot be excluded in *E. coli*. Available commercial systems, such as the AmpC gradient test, Rosco Tablets or MAST “AmpC detection disk set”, provide suboptimal performance and/or time-to-result [14].

Rapid phenotypic antimicrobial susceptibility assays are needed, and a disruptive technology based on flow cytometry has been proven with results showing it to be an excellent tool [15]. Instead of looking for their ability to grow in the presence of the drugs, the cell lesions are detected after short periods of incubation. Moreover, it could help with understanding the underlying resistance mechanisms such as ESBL [16] or carbapenemases [17]. FASTinov already validated a FASTgramneg kit that screens for AmpC in a two-hour assay. In this article, we describe a rapid flow cytometry assay which is able to detect the presence of pAmpCs on strains that screened positive in a maximum of 2 h in the *Enterobacterales* group-I strains. The technology, developed by FASTinov, a spin-off of Porto University, Portugal, is already granted in EUA, Europe, India, Japan and Brazil.

## 2. Results

In this paper, a rapid flow cytometry assay for AmpC detection was compared with the reference method (Figure 1); both are phenotypic inhibitor-based methods. The phenotypic characterization of the strains, according to the reference method, showed that 42 strains were classified as pAmpC producers (22 tested at FASTinov and 20 tested at Ramón y Cajal University Hospital). Some of these strains produced ESBLs (*n* = 10) and carbapenemases (*n* = 1), or both (*n* = 1). Nineteen percent of the isolates (19/100) showed the production of more than one type of β-lactamase. Moreover, forty-two isolates (42/100) were carbapenemase producers (*n* = 23 KPC, *n* = 10 metallo-carbapenemases and *n* = 9 OXA-48-like). Note that 9% of the tested isolates (9/100) were negative for the production of ESBLs, carbapenemases or pAmpC beta-lactamases despite resistance to cefoxitin and cefotaxime and/or ceftazidime. Overall, the results of pAmpC detection as well as other β-lactamases are detailed in Figure 2.

The flow cytometry results, after the definition of cut-off values, were calculated using the proprietary software (bioFast, FASTinov, Porto, Portugal). A report was then produced for each strain, with the flow cytometry expertise of the user not a necessary requirement. The new FASTinov flow cytometry assay for pAmpC detection showed a percentage of agreement of 93.0% with the reference method and a sensitivity of 95.5% and a specificity of 91.1%. Two false-negative isolates (*K. pneumoniae* and *Proteus mirabilis*) were found after comparison with the disk diffusion reference method. Moreover, the flow cytometry assay showed five false-positive results.

Molecular assays were performed on discrepant results. The two strains that were negative after the flow cytometry assay and positive when using the reference method were true false negatives as the molecular assay confirmed that they were positive pAmpC (DHA) producers. Five discrepant results that were positive following the flow cytometry assay and negative when using the reference method were also true false-positive strains as they were negative for pAmpC production when using the molecular assay; all of them were carbapenemase-producing *K. pneumoniae* (four KPC and one OXA-48). 

An example of the flow cytometry analysis of the pAmpC-producer and non-producer isolates is shown in Figure 3. Both strains are resistant to cephalosporins, so a low intensity of fluorescence was observed when the strains were incubated with the cephalosporin alone; the fluorescence of the cells was similar to the non-treated cells (A or B). An evident increase in the intensity of fluorescence was observed after incubation of 1 h with cloxacillin together with the cephalosporins (cefotaxime or ceftazidime) on the pAmpC-positive strain, but not on the negative one (see Figure 4 for an example).

## 3. Discussion

pAmpC β-lactamases have been described worldwide. A study performed in Portugal showed a decrease in the prevalence of the DHA-1 enzyme (55%), despite it being the most detected enzyme, and this result is still evident in the research from different institutions. An increase in the detection of the CMY-2 enzyme (44%) reinforces the fact that epidemiological changes occur [18]. On the other hand, in 2000 in Spain, Bou et al. described the first pAmpC present on a FOX-447 *E. coli* producer [19]. Later, a Spanish revision performed in 2012 showed that the national prevalence of pAmpCs was about 0.6%, with CMY-2 (66.7%) being the most represented, followed by DHA-1 (25.6%) [8]. These studies include molecular methods for the detection of these enzymes. These molecular assays could be a rapid alternative to phenotypic tests, but several genes need to be screened, and discrepancies between molecular and phenotypic assays have also been reported [20]. MALDI-TOF MS is also able to detect pAmpC enzymes after 3–4 h incubation with different drugs [21,22,23], but optimization is still needed.

The most widespread phenotypic method among clinical microbiology laboratories to detect pAmpCs is related to double disc synergy and the combination with inhibitors due to their low cost and effectiveness. Other methods such as the 3D test and/or Etest (cefotetan/cefotetan + cloxacillin) have also been proposed [9]. Despite their higher sensitivity and specificity (~85% and 94% respectively), they are more expensive and complex to use. In addition, the chromogenic cica-beta test has been shown to have a low sensitivity (<20%) for pAmpC detection [9,14].

Here, we describe a new, rapid and inhibitors-based assay to detect pAmpC enzymes by flow cytometry technology. Flow cytometry has proved to be an accurate and rapid technique for the evaluation of the antimicrobial susceptibility profiles of bacteria both in human [15] and veterinary [24] fields. Flow cytometry also detects the relevant mechanisms of resistance such as ESBLs or carbapenemases [16]. After a short incubation time with different antimicrobials and the in-house optimized fluorescent probes, several cellular parameters such as cell size, complexity and intensity of fluorescence are evaluated and correlated with the susceptible phenotype.

The new flow cytometry assay described here is based on EUCAST guidelines, i.e., it compares the effect of the two cephalosporins—cefotaxime and ceftazidime—alone and in combination with an AmpC inhibitor such as cloxacillin [13]. Because it is a culture-independent method, the time-to-result is much lower (~2 h) than any other phenotypic assay currently available on the market. The bacterial cell lesions produced by the drugs are recorded by the flow cytometer after staining with a fluorescent probe. A synergic effect was observed when cephalosporins associated with the inhibitor (cloxacillin) showed an increase in intensity of fluorescence compared to the cephalosporin alone. The new proposed flow cytometric assay for pAmpCs was able to correctly detect 93% of the cases. False-positive results were found in the isolates with concomitant carbapenemases, especially KPCs, probably due to the similarities in the inhibition profiles.

The work described here was performed from pure colonies after sub-culturing in broth media to be in an exponential growth phase, as previously described [15]. Nevertheless, it has the potential to be used directly on positive blood cultures [15].

A limitation of this work could be the limited number of isolates, especially those presenting the concomitant mechanisms of resistance and the absence of *E. coli* with the constitutive expression of the chromosomal AmpC enzyme that might resemble a pAmpC producer in phenotypic assays. 

Rapid antimicrobial susceptibility testing contributes to better antimicrobial use and it has been advocated in antimicrobial stewardship programs. Additionally, it contributes to avoiding the spread of multidrug-resistant microorganisms, including pAmpC producers that have been responsible for several outbreaks worldwide [25,26]. Moreover, the understanding of the most common resistance mechanisms is relevant to the design of new strategies, including new drugs, or the design of enzyme inhibitors that could recover the antimicrobial activity.

## 4. Materials and Methods

Isolates: One hundred non-duplicated Gram-negative bacilli, well-characterized bacteria, were tested (forty-two from FASTinov; twenty-seven from CCP—Culture Collection of Porto—Faculty of Pharmacy University of Porto [18]; thirty-one from Microbiology Service of Ramón y Cajal Hospital, Madrid, Spain). Regarding species identification, 74 were *K. pneumoniae*, 19 were *E. coli*, 3 were *P. mirabilis*, 2 were *Klebsiella aerogenes*, 1 was *Klebsiella variicola* and 1 was *Kluyvera cryoscrescens* provided from different sources (blood cultures, urine, bronquial secretions). All of them were resistant to ceftazidime and/or cefotaxime and resistant to cefoxitin. In total, 9 strains were negative, 42 were AmpC positive and 49 presented other mechanisms of resistance. Two controls were added: *E. coli* ATCC 25922 as a negative control and a well-characterized *E. coli* CMY-17 as a positive control [27].

### 4.1. Detection of pAmpC β-Lactamases

AmpC production was phenotypically confirmed by disk diffusion with “AmpC Confirm Kit: AmpC Enterobacteriaceae” (Rosco Diagnostica, Taastrup, Denmark), and this was considered the reference method. Moreover, a new flow cytometry assay was performed using the same reasoning (synergic studies with inhibitors) in two different laboratories: the FASTinov in Porto (FASTinov + FFUP bacterial collections) and the Hospital Ramón y Cajal in Madrid (Ramón y Cajal collection). Briefly, overnight cultures in blood agar plates (bioMérieux, Marcy-l’Étoile, France) were used to prepare a bacterial suspension in Brain Heart broth (Sigma-Aldrich, St. Louis, MO, USA) incubated with shaking until turbidity (around 1–1.5 h). After that time, a centrifugation step was performed and the suspension adjusted to 0.5 McFarland and diluted (1:100) in filtered Mueller–Hinton II Broth which was cation-adjusted (Becton Dickinson, Franklin Lakes, NJ, USA). Each cell suspension (100 µL) was inoculated in a 96-well microplate panel previously prepared with 2 µg/mL of cefotaxime or 4 µg/mL of ceftazidime alone or associated with 500 µg/mL of cloxacillin; a membrane depolarization fluorescent dye was added to each well. After a 1-hour incubation at 35 ± 2 °C with shaking and protection from light, the panel was analyzed by flow cytometry using the CytoFLEX (Beckman Coulter, Brea, CA, USA) flow cytometer. Cell fluorescence intensity, the number of events, and the morphological changes were recorded in an FSC (flow cytometer standard) file using bioFAST software (SW). This software includes a proprietary algorithm, uses FCS files provided by the flow cytometer and then automatically reports each strain as positive or negative for pAmpCs. The SW analysis was performed independently in both sites (Porto and Madrid). The number of events, the light scatter values and the fluorescence intensity of each well were incorporated in the algorithm. Cut-off values for the flow cytometry assay were calculated using ROC curves and introduced on the bioFAST SW. Discrepant results regarding pAmpC production between the flow cytometry assay and the reference method, as well as the negative results on the negative pAmpC enzymes, were confirmed by a multiplex PCR using six sets of primers able to detect the most relevant plasmid-mediated AmpC families: CIT, DHA, ACC, FOX, MOX and EBC [28].

### 4.2. Detection of Other β-Lactamases

The production of carbapenemases and/or ESBLs on the studied isolates was also phenotypically screened using the disk diffusion assay KPC, MBL and OXA-48 confirm kit: carbapenemases and Total ESBL Confirm Kit: ESBLs (Rosco Diagnostica, Taastrup, Denmark).

### 4.3. Statistical Analysis

(1) Determination of pAmpC by flow cytometry assay: The cut-off values for flow cytometry were calculated using ROC curves. The cut-off values were introduced into the software developed by FASTinov, in order to detect the presence of pAmpCs according to EUCAST [13]. (2) Comparison with the reference method: The percentage of agreement as well as the sensitivity (false negatives) and specificity (false positives) of the performance of the FASTinov method were calculated according to ISO 20776-2:2021 [29] using disk diffusion as the reference method.

## 5. Conclusion

Flow cytometry has been an outstanding tool in several medical fields and could now be used in microbiology diagnosis. It provides a rapid susceptibility phenotype and clarifies the main mechanism of resistance, such as the presence of AmpCs, with good correlation with the reference phenotypic method, despite its known limitations. Moreover, it will provide important and rapid information to the clinicians. This new assay will contribute to a faster and more accurate treatment of patients and, if necessary, its isolation can help to prevent the spread of the mechanism of resistance.

## 6. Patents

1.Kit and method for detecting the resistant microorganisms to a therapeutic agent. WO/2012/164547 A1. Granted in Europe (Application granted 6 December 2012).2.Fluorescent method for detecting microorganisms resistant to a therapeutic agent. US14123757. Granted in USA (Application granted 22 March 2016).

## Figures and Tables

**Figure 1 antibiotics-11-01130-f001:**
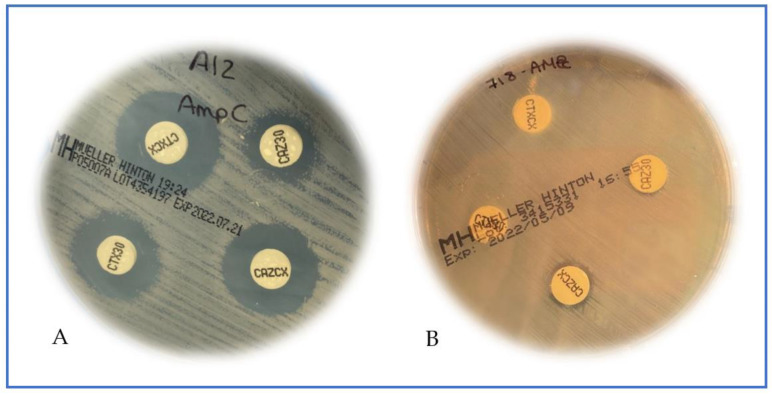
Phenotypic inhibitor-based methods (disk diffusion); (**A**): pAmpC positive, difference of ≥5 mm between one or both cephalosporins when comparing with the inhibitor tablet; (**B**): pAmpC negative, no differences between cephalosporins disks and corresponding inhibitors.

**Figure 2 antibiotics-11-01130-f002:**
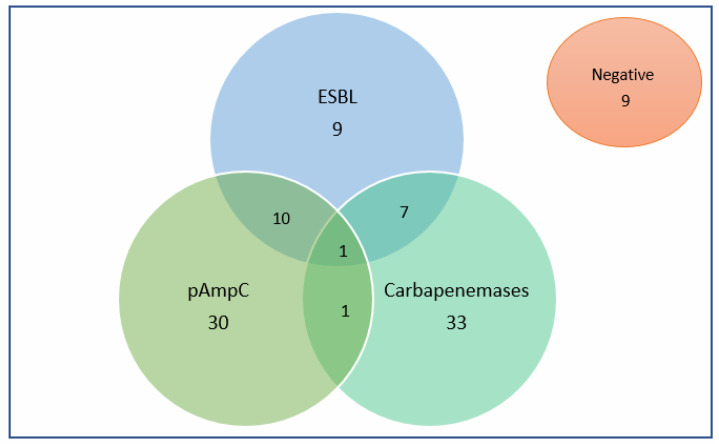
Phenotypic characterization of the strains by reference methods according to the ability to produce β-lactamases.

**Figure 3 antibiotics-11-01130-f003:**
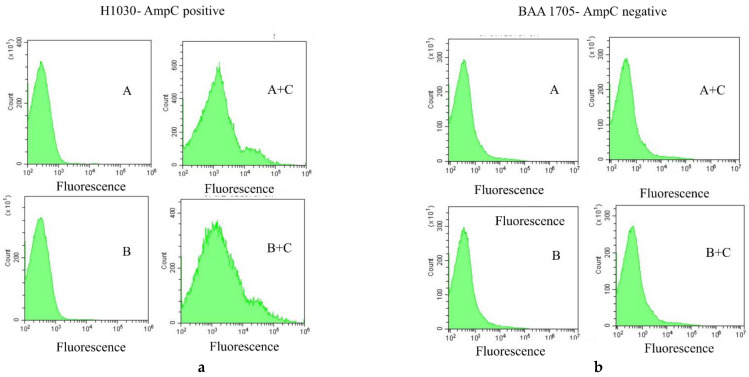
Histograms representing an example of one pAmpC-positive (H1030) and one pAmpC-negative (BAA 1705) strain. (**a**) Cells incubated for 1 h with cefotaxime; (**b**) Cells incubated for 1 h with ceftazidime; (A+C) and (B+C) Cells incubated with A or B associated with cloxacillin (C) for 1 h. An increase in the intensity of fluorescence (peak in the histogram shift to the right) is evident only on the pAmpC-positive strain when cloxacillin is associated with the cephalosporins.

**Figure 4 antibiotics-11-01130-f004:**
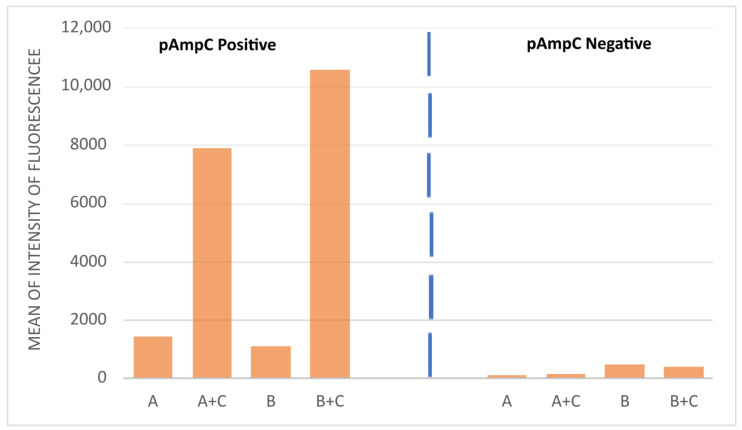
Bar chart representing the intensity of fluorescence of the cells of an example of a pAmpC-positive (strain H1030) and a pAmpC-negative (strain BAA 1705) strain. (A) Cells incubated for 1 h with cefotaxime; (B) Cells incubated for 1 h with ceftazidime; (A+C) and (B+C) Cells incubated with cefotaxime (A) or ceftazidime (B) associated with cloxacillin (C) for 1 h. The increase in the intensity of fluorescence means cell lesion, i.e., susceptibility. Both strains are resistant to both cephalosporins; an increase in the intensity of fluorescence was evident only on the pAmpC-positive strain when cloxacillin is added to cephalosporins.

## Data Availability

Data is contained within this article.

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
