# Peer review of "Rapid Detection of Plasmid AmpC Beta-Lactamases by a Flow Cytometry Assay"

_antibiotics, 2022, doi:10.3390/antibiotics11081130_

Round 1

Reviewer 1 Report

Comments to the authors a for evaluating the following manuscript

Title:  “A new method for rapid detection of plasmid AmpC beta-lactamases based on flow cytometry”dear authors; 

 The aim of the manuscript is interesting, the overall of this manuscript were well written; meanwhile, there are several criticisms listed below you have to manage them 

Title: the title should be rephrased I suggest the following title “ Rapid detection of plasmid AmpC beta-lactamases by flow cytometry assay”

Keywords: Key words of the manuscript are too long and uninformative, you must concise them.

Abstract section

·       Please add the overall practical implementation of important of rapid detection of pAmpC enzymes

Introduction section

·       The first part of the literature is quite «old» “1980;2013” and insufficient. Please add literature of the last five years. There is some bibliography placed over the last five years?

·       The information in the introduction section were insufficient please added more details

·       Please if you can elaborate in detail the necessary for rapid detection of plasmid AmpC beta-lactamases in the clinical fields line 54;57

·       you have to add a paragraph about the spreading of the resistant pathogens “ bacteria and fungi “ and the public panic from this crises using the following reports “The antimicrobial resistance crisis: causes, consequences, and management”; ;“Characterization of Methicillin Resistant Staphylococcus aureus isolated from human and animal samples in Egypt” ; “Alternative therapy to manage otitis media caused by multidrug-resistant fungi”

Results

·       The design of tables is very old, kindly change this design

·       Can you added a dendrogram or figure for all tested isolates to show the percentage of positive and negative rsults

·       You have to add the sensitivity and specificity of the new method compare to the phenotypic and genotypic methods which utilized in this study

Discussion:

·       Can you add a paragraph about the using of flow cytometer for rapid detection of resistant pathogens as in “A Rapid Flow Cytometric Antimicrobial Susceptibility Assay (FASTvet) for Veterinary Use – Preliminary Data”

Methodology

·       Please add the source of isolates, positive and negative control strains

Conclusion section:

·       Please add a conclusion section, it is very important in any report

Author Response

Title: the title should be rephrased I suggest the following title “Rapid detection of plasmid AmpC beta-lactamases by flow cytometry assay”.

Authors: yes, we agree with this modification and change was made.

Keywords: Key words of the manuscript are too long and uninformative, you must concise them.

Authors: we have already changed the keywords in this new version of manuscript, we believe they are now more informative and shorter than previous ones as recommended.

Abstract section: Please add the overall practical implementation of important of rapid detection of pAmpC enzymes

Authors: We have added the importance of the rapid detection of pAmpC such as “for patient treatment and also for epidemiological purposes”

Introduction section

  • The first part of the literature is quite «old» “1980;2013” and insufficient. Please add literature of the last five years. There is some bibliography placed over the last five years?

Authors: We have included now more recent references.

  • The information in the introduction section were insufficient please added more details
  • Please if you can elaboratein detail the necessary for rapid detection of plasmid AmpC beta-lactamases in the clinical fields line 54;57
  • you have to add a paragraph about the spreading of the resistant pathogens “ bacteria and fungi “ and the public panic from this crises using the following reports “The antimicrobial resistance crisis: causes, consequences, and management”; ;“Characterization of Methicillin Resistant Staphylococcus aureusisolated from human and animal samples in Egypt” ; “Alternative therapy to manage otitis media caused by multidrug-resistant fungi”     

Authors: Thanks for the suggestions, we have rewrite part of the introduction.

Results

  • The design of tables is very old, kindly change this design

Authors: The tables were removed and 2 new fig included

  • Can you added a dendrogram or figure for all tested isolates to show the percentage of positive and negative results

Authors: A Venn diagram was added.

  • You have to add the sensitivity and specificity of the new method compare to the phenotypic and genotypic methods which utilized in this study

Authors: The sensitivity and specificity of the method was calculated using as reference the phenotypic tests as this new method is an inhibitor based method. The genotypic method was only used on discrepant results.

Discussion

  • Can you add a paragraph about the using of flow cytometer for rapid detection of resistant pathogens as in “A Rapid Flow Cytometric Antimicrobial Susceptibility Assay (FASTvet) for Veterinary Use – Preliminary Data”

Authors: We added a sentence on the Discussion section

Reviewer 2 Report

The authors aimed in this work to detect the plasmid harboring resistance AmpC beta-lactamases using flow cytometry. 

1- However, with my interest to read and know the idea behind this work, I am asking what is the real benefit behind this work, especially in the practical field. Will the clinicians or researchers use flow cytometry to just detect the resistance gene which can be detected much easier and cheaper by other simple methods? Even the authors did not describe this issue in the introduction or explain what are the benefits.

2- The manuscript is scanty and lacks a lot of knowledge that may be helpful for readers, they have to explain their findings in flow cytometry in detail interrupting the figures for readers to be fully understood.

3-   There is no argument about the importance of flow cytometry, but it is still expensive and not understood by many coworkers, and its interpretation needs experts. At least draw attention to how did you interrupt these results.

I think this work requires major modifications to be accepted. 

Author Response

Methodology

  • Please add the source of isolates, positive and negative control strains.

Authors: The source of the strains was added; the controls belong to ATCC strains. Number of these controls are also included.

Conclusion section

  • Please add a conclusion section, it is very important in any report

Authors: A conclusion section was added to the manuscript

The authors aimed in this work to detect the plasmid harboring resistance AmpC beta-lactamases using flow cytometry. 

1-However, with my interest to read and know the idea behind this work, I am asking what is the real benefit behind this work, especially in the practical field. Will the clinicians or researchers use flow cytometry to just detect the resistance gene which can be detected much easier and cheaper by other simple methods? Even the authors did not describe this issue in the introduction or explain what are the benefits.

Authors: Flow cytometry has been demonstrated as a very promising solution for rapid antimicrobial susceptibility testing. Laboratories implementing this technology could not only determine if the isolate is S or R but also to decipher the presence of some resistance mechanisms such as ESBLs or carbapenemases and now AmpC beta-lactamases. Moreover, it has also the advantage of detecting relevant from epidemiological purpose. Those benefits are now better explained on the paper.

2- The manuscript is scanty and lacks a lot of knowledge that may be helpful for readers, they have to explain their findings in flow cytometry in detail interrupting the figures for readers to be fully understood.

There is no argument about the importance of flow cytometry, but it is still expensive and not understood by many coworkers, and its interpretation needs experts. At least draw attention to how did you interrupt these results.

Authors: We agree with the referee`s views. In fact, a software was developed in order to give the answer about the presence of AmpC without a lot of expertise on flow cytometry. This information is now added to the paper. We believed the figures are now better explained.

3- I think this work requires major modifications to be accepted. 

Authors: We hope that modifications introduced in the manuscript gives a positive view of the referee.

Reviewer 3 Report

Manuscript Number antibiotics-1821836
"A new method for rapid detection of plasmid AmpC beta-lactamases based on flow 2 cytometry"

The paper provides a simple and novel method for rapid detection of plasmid AmpC beta-lactamases using flow cytometry. The paper is well written and has clinical applications. There are few minor revisions.

Please find below my comments:

-        Change the 1st sentence to - but they are not routinely investigated in clinical laboratories (line no. 22 and 23)

-        Reframe line no. 56 and 57

-        Have the authors performed statistical analysis for flow cytometry studies? Please mention in the manuscript.

-        The flow cytometry results should be represented in a bar chart form apart from showing the histograms.

-        It would be good if the authors can include the data and images of disk diffusion assay (AmpC) performed in the current study.

-        General comments for results section – please provide 1-2 lines of introduction (when starting a paragraph) about need for the study in each section of results.

-        Conclusion section is not provided under separate heading.

-        Recheck for minor grammatical errors and English language at few places.

Author Response

  • Change the 1stsentence to - but they are not routinely investigated in clinical laboratories (line no. 22 and 23)

Authors: we made this modification

  • Reframe line no. 56 and 57

Authors: we made this modification

  • Have the authors performed statistical analysis for flow cytometry studies? Please mention in the manuscript.

Authors: Statistical analysis was performed for flow cytometry data analysis. This      information was included in the new version of the manuscript

  • The flow cytometry results should be represented in a bar chart form apart from showing the histograms.

Authors: We have added it. Thanks for the suggestion

  • It would be good if the authors can include the data and images of disk diffusion assay (AmpC) performed in the current study.

Authors: Images added of the performance of disk diffusion method

  • General comments for results section – please provide 1-2 lines of introduction (when starting a paragraph) about need for the study in each section of results.

 Authors: This is now added in the new version of the manuscript.

  • Conclusion section is not provided under separate heading.

   Authors: Conclusion is now under a separate heading

-        Recheck for minor grammatical errors and English language at few place

   Authors: We have reviewed the error grammars

Reviewer 4 Report

This manuscript by Ines Oliveira et al. describe a new method for detection of plasmid AmpC betalactamase based on flow cytometry.

This manuscript deserves major revision before possible acceptance for publication.

Global: prefer passive expressions

The results would be more interpretable as a Venn Diagram

Numbers less than 12 must be written in full.

Table 1: the determination of pAmpC by RM is not interesting because RM is considered as a gold standard, its performance must be 100%.

Table 2: What do the authors mean: K098n EB034, 081, R68, R66, P51, V28? Authors should explain the acronyms.

"ie.", "et al" should be written in italics.

The conclusion is not accurate because the methods described could not possibly be of value in treating patients (and because the authors did not provide information about this possible value...).

Methods: How was the number of strains to be included determined? What were the selection criteria? How did the authors choose between cefotaxime and ceftazidime (line 200).

The reference methods are quite surprising as molecular biology is becoming more and more efficient.

Author Response

Global: prefer passive expressions

Authors: English was revised.

The results would be more interpretable as a Venn Diagram

Authors: A Venn diagram is now included

Numbers less than 12 must be written in full.

Authors: It was corrected in the new version

Table 1: the determination of pAmpC by RM is not interesting because RM is considered as a gold standard, its performance must be 100%.

Authors: Thanks for noting this point.

Table 2: What do the authors mean: K098n EB034, 081, R68, R66, P51, V28? Authors should explain the acronyms.

Authors: There are coded for strains but we decided to eliminate them as they do not add additional information.

"ie.", "et al" should be written in italics.

Authors: This has been modified.

The conclusion is not accurate because the methods described could not possibly be of value in treating patients (and because the authors did not provide information about this possible value...).

Authors: Conclusion is now in a separate heading. 

Methods: How was the number of strains to be included determined? What were the selection criteria? How did the authors choose between cefotaxime and ceftazidime (line 200).

Authors: The criteria of selection of strains was to include well characterized strains that we had access. We used both cephalosporins which agrees with EUCAST protocol of Mechanisms of resistance https://www.eucast.org/resistance_mechanisms/

The reference methods are quite surprising as molecular biology is becoming more and more efficient.

Authors: We agree with the referee´s views. However, we decided to use the phenotypic test as this is more accessible to the clinical laboratories at present and because flow cytometry is also a phenotypic test. With this comparison we avoid also the eventually presence of a resistance determinant (gene) and no expression of it.

Round 2

Reviewer 1 Report

thank you for your effort to manage my criticisms in this new versions

Author Response

Thank you for your advices and appreciate the changes in the latest version. In this new actualization we have tried to review English language style and minor changes have done.

Reviewer 4 Report

Except for the two last point of my comments (that could not be adressed if I understand the answer of the authors), all my previous comments have lead to revision.

Author Response

Thank you for your suggestions. We have made some modifications in the introduction and a new reference was added. Minor changes in all the manuscript have been performed too.